# A multi-omics analysis unveils functional and regulatory links between hydroxybenzene and aromatic amino acid metabolism in *Candida albicans*

Enrico Garbe,[1] Sascha Schäuble,[2] Bettina Böttcher,[1] Robert Jesse,[1] Dominik Driesch,[3] Lasse van Wilijck,[4] Osama Elshafee,[5] Tim Bastian Schille,[5,6] Bernhard Hube,[5,6,7] Gianni Panagiotou,[2,6,7,8] Slavena Vylkova[1,9]

**ABSTRACT**   The fungus *Candida albicans* is a frequent colonizer of humans but also an opportunistic pathogen causing superficial to severe infections, especially in vulnerable individuals. Its broad metabolic flexibility is key for the fungal adaptation to host environments, evasion from immune attack, and virulence. Amino acid metabolism and homeostasis, in particular, are critical for fungal fitness—illustrated by a rapid metabolic shift in response to amino acid starvation to restore intracellular metabolic balance. To investigate the cellular mechanisms underlying such compensatory metabolic processes, we performed data-driven genome-scale metabolic modeling based on transcriptional metabolic profiles of amino acid-starved cells to identify condition-specific fungal metabolic fluxes and pathway activities specific to cellular response to amino acid starvation. Most prominently, we predicted altered activity of the shikimate pathway upon amino acid limitation and identified a simultaneous induction of aromatic amino acid (AAA) biosynthesis and a metabolic gene cluster required for the catabolism of hydroxybenzenes. Further phenotypic and transcriptional analyses not only verified the transcription factor Zcf25 as the central regulator of the catechol-branch of this pathway, but also condition-specific co-regulation of AAA and hydroxybenzene metabolism mediated by Zcf25 and the transcriptional regulator of amino acid metabolism Stp2. These findings propose a so far unknown metabolic link between amino acid and hydroxybenzene metabolism in *C. albicans*, therewith adding another layer to its metabolic plasticity.

**IMPORTANCE**   The opportunistic human fungal pathogen *Candida albicans* possesses a remarkable metabolic plasticity, which is essential for both fungal commensalism and virulence and influences its physiology and behavior in multiple ways. The investigation of such processes particularly benefits from the emergence of multi-omics and *in silico* approaches. In this study, we combined a multi-omics approach with genome-scale metabolic modeling to investigate the fungal metabolic adaptation to amino acid utilization and starvation. Most strikingly, we found an altered activity of the shikimate pathway upon amino acid starvation, accompanied by a simultaneous induction of two metabolic gene clusters required for the metabolism of hydroxybenzenes. Further analyses revealed so far unknown potential functional and regulatory links between both metabolic pathways, which provide starting points for future research leading to a better understanding of the fungal adaptation to dynamic host conditions.

**KEYWORDS**   *Candida albicans*, metabolomics, shikimate pathway, 3-oxoadipate pathway, amino acid metabolism, genome-scale metabolic modeling

**Peer Reviewer** Jozef Nosek, Comenius University, Bratislava, Slovak Republic

Address correspondence to Enrico Garbe, enrico.garbe@leibniz-hki.de, or Sascha Schäuble, sascha.schaeuble@leibniz-hki.de.

Enrico Garbe and Sascha Schäuble contributed equally to this article. The author order was determined based on their contributions to the article.

The authors declare no conflict of interest.

See the funding table on p. 17.

10.1128/msystems.00226-25 **1**

*C*andida albicans is a human commensal fungus, able to cause superficial to life-threatening systemic infections in susceptible individuals (1, 2). In its commensal stage, the fungus typically colonizes the mucosal surfaces of the gastrointestinal and urogenital tract (3). In these niches, the fungus is constantly challenged by dynamic environmental conditions, such as limited or changing nutrient availability (4, 5). The ability to quickly adapt to its environment is thus pivotal for *C. albicans* and leveraged by a remarkable metabolic plasticity, enabling it to utilize a plethora of nutrient sources (6, 7). Metabolic stimuli and traits further shape the fungal persistence and virulence in multiple ways, for example, by initiation of filamentation or stress responses (8–12).

Amino acid metabolism, in particular, plays a central role in this context. Ubiquitously available in the human host (either free or peptide-bound), certain amino acids are utilized exceptionally well as nitrogen and carbon sources by *C. albicans* (13). Although the fungus is able to synthesize all proteinogenic amino acids on demand by itself, it prefers the energetically more favorable uptake of external amino acids (13). For this, it deploys sets of specialized enzymes and transporters, including (oligo)peptide transporters, secreted proteases for liberation of peptide-bound amino acids, and a family of amino acid permeases (AAPs) with varying substrate specificities for their uptake (14–20). The sensing and uptake of extracellular amino acids is also closely intertwined with the fungal virulence and host cell interaction, for example, fungal resistance against immune cell attack or the amino acid-induced yeast-to-hyphae transition (14, 17, 21–25).

The key regulator of the utilization of extracellular amino acids is the S̲sy1-P̲tr3-S̲sy5 (SPS) sensor system with its downstream effector—the transcription factor Stp2, which mediates the expression of SPS target genes, like AAPs, in response to extracellular amino acid availability (26–28). Inactive Stp2 is retained in the cytoplasm and undergoes endoproteolytic cleavage immediately upon recognition of extracellular amino acids, allowing translocation into the nucleus and rapid activation of transcriptional responses (26). Consistent with the emphasized importance of amino acid utilization for fungal virulence, deletion strains of SPS sensor components and particularly Stp2 are impaired in biofilm formation, fungal resistance against macrophage attack, and stress implied by reactive oxygen species (17, 21–23, 29). Furthermore, we previously found a remarkable metabolic flexibility in amino acid-starved *stp2*Δ biofilms, allowing them to re-establish wild-type (WT)-like metabolic homeostasis upon prolonged incubation, pointing to compensatory mechanisms alleviating amino acid starvation (30). However, the full extent of Stp2-mediated amino acid uptake and maintenance, its underlying mechanisms, and how this modulates pathogenic and commensal behavior in the host remains unclear.

In this study, we analyzed the metabolic rearrangements in *C. albicans* during amino acid utilization and starvation, using a multi-omics-driven combined experimental-computational approach. We generated transcriptional and metabolic profiles of *C. albicans* WT and *stp2*Δ cells in media with or without excess amino acids and combined them with genome-scale modeling of fungal metabolism. Our analyses reveal prominent metabolic flux differences in aromatic amino acid (AAA) metabolism in amino acid-starved *stp2*Δ cells and show that this is accompanied by the upregulation of a 3-oxoadipate pathway-associated metabolic gene cluster (MGC) required for the catabolism of hydroxybenzenes (31), and verify the transcription factor Zcf25 as the main regulator of this MGC. It further provides evidence that a condition-specific hitherto unknown cross-regulation between AAA and hydroxybenzene metabolism, mediated by Stp2 and Zcf25, exists. Taken together, our study suggests a novel link between both the AAA and hydroxybenzene metabolic circuits, potentially involved in the fungal adaptation to changes in amino acid availability, and thereby contributes to the growing understanding of metabolic adaptation of this opportunistic pathogen.

## RESULTS

### Deletion of Stp2 results in global metabolic changes and delayed response to extracellular amino acids

We first investigated the metabolic changes in the fungus in response to amino acid availability. We performed global metabolomic analysis to obtain cellular metabolite profiles of *C. albicans* WT and *stp2Δ* cells incubated for 90 min in Stp2 non-inducing (SD; glucose-rich and amino acid-free) and inducing (SC [glucose-free and amino acid-rich] and CAA [glucose-free, amino acid and peptide-rich]) media (27, 28, 30). We also obtained metabolic profiles from WT and *stp2Δ* cells after prolonged incubation (8 h) in CAA, where *stp2Δ* growth is slower but present, to examine the processes leading to the re-establishment of metabolic homeostasis in *stp2Δ* (23).

A total of 586 unique metabolites were identified and unambiguously assigned to one super and one sub pathway, which represent specific parts of the super pathways (e.g., arginine biosynthesis as a part of the amino acids super pathway) (Table S1). Most metabolites were detected from the super pathways' amino acids and lipids, with 183 and 176 metabolites, respectively (Fig. 1A; Table S1). A principal component analysis (PCA) revealed predominantly distinct clusters for the replicates of the individual sample groups (Fig. 1B). Furthermore, we noted a clear separation between the samples from amino acid-free SD medium and amino acid-rich SC and CAA media for both WT and *stp2Δ*. While WT and *stp2Δ* 90 min samples clustered together for SD medium, they displayed separation for SC and CAA media, indicating metabolic differences. Furthermore, for both WT and *stp2Δ*, the 90 min samples for SD and CAA formed clusters that were distinct from the corresponding 8 h samples, indicating substantial metabolic changes in both strains upon prolonged incubation (Fig. 1B).

We next analyzed the abundance profiles of distinct metabolites between the amino acid-rich medium (SC and CAA) and the amino acid-free medium (SD) in the WT. About three-fourths of all detected metabolites were significantly changed in abundance in either SC and CAA or both compared to SD (Fig. 1C). While we observed 166 metabolites with higher and 92 with lower abundances in both media, we also found additional medium-specific effects (Fig. 1C). The metabolites with medium-specific differences were assigned to a broad variety of diverse sub-pathways, including a notable enrichment for amino acid metabolic pathways for SC as well as lipid- and peptide-associated pathways for CAA medium (Table S1). Upon prolonged incubation in amino acid-rich CAA medium, the WT still showed a substantial number of metabolites with increased or lowered abundance compared to SD medium, which partially overlapped with the early response (Fig. S1A). Most abundant at both time points were intermediates from the metabolism of branched-chain amino acids (BCAA), histidine, cysteine, and methionine (Table S1). Overall, metabolites from each super pathway were present in similar numbers in the early and late-stage WT response to amino acid-rich CAA medium, except for amino acids, nucleotides, and lipids, which showed more enrichment at 90 min. Interestingly, lipids also showed increased metabolite depletion at 8 h (Fig. S1B and Table S1).

Since accumulation or depletion of intermediates associated with a given pathway can be indicative of reduced or increased metabolic flux levels and pathway activity (32, 33), we conducted a KEGG pathway enrichment analysis to identify potential differences in pathway activity by comparing metabolite abundance profiles for the WT from amino acid-rich media vs. amino acid-free medium (34). Overall, the results were similar for both amino acid-rich media types, CAA and SC (Fig. 1D; Fig. S1C). Next to vitamin and nucleotide metabolism, we identified, based on more abundant metabolites, several enriched pathways related to amino acid metabolism, such as for cysteine, proline, and aspartate, as well as valine, leucine, and isoleucine biosynthesis, likely related to the increased availability of amino acids in the amino acid-rich medium (Fig. 1D; Fig. S1C). In contrast, pathways related to sugar and central carbon metabolism, including amino sugar and pyruvate metabolism or glycolysis/gluconeogenesis, were enriched based on

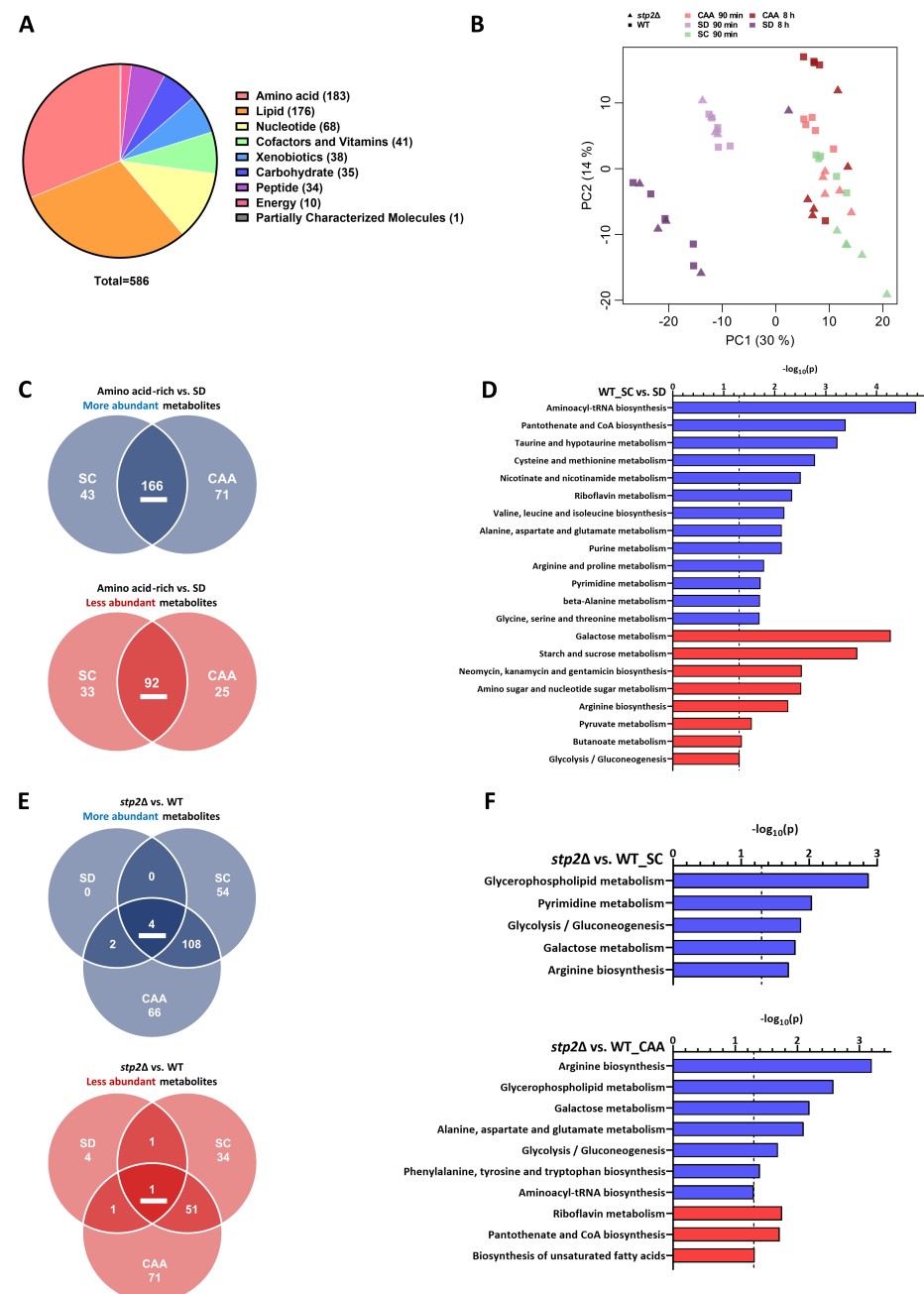

**FIG 1** Metabolic profiling of WT and *stp2* Δ in amino acid-free and -rich medium. (A) Number of metabolites detected in this study, assigned to their respective "Super pathways." (B) PCA of WT and *stp2*Δ metabolic profiles obtained from the indicated conditions. (C) Metabolites with significantly increased or lowered abundance (*P* < 0.05) in amino acid-rich media compared to SD in the WT (t = 90 min). (D) Metabolites either significantly more (blue) or less (red) abundant in SC vs. SD in the WT possessing a unique HMDB identifier were used for KEGG pathway enrichment analysis via "MetaboAnalyst." Dashed line indicates the significance threshold of *P* = −log10(0.05). (E) Metabolites with significantly increased or lowered abundance (*P* < 0.05) in *stp2*Δ vs. WT (t = 90 min). (F) KEGG pathway enrichment analysis via "MetaboAnalyst" for *stp2*Δ vs. WT in CAA and SC with either significantly more (blue) or less (red) abundant metabolites.

less abundant metabolites suggesting a potential need for higher metabolic activity in these pathways (Fig. 1D; Fig. S1C).

When comparing metabolite abundances between *stp2*Δ and the WT next, as expected, only minor differences were detected in the Stp2-non-activating medium SD, while a considerable number of metabolites showed differential abundance in

both amino acid-rich media (Fig. 1E). Notably, *stp2Δ* generally showed a trend toward metabolite accumulation rather than depletion, except for metabolites associated with the lipid super pathway which showed the opposite tendency (Fig. S1D). During prolonged incubation (8 h), *stp2Δ* showed a substantially lower amount of differentially abundant metabolites in CAA vs. SD than the WT (Fig. S1A and E). Yet, no significant differences in metabolite abundance were found for *stp2Δ* vs. WT after 8 h incubation in both SD and CAA, suggesting that *stp2Δ* is capable of adapting its metabolism to amino acid-rich conditions over time despite its impaired amino acid uptake (Table S1). In line with this notion, the enrichment analysis based on more abundant metabolites for *stp2Δ* vs. WT in amino acid-rich media associated pathways such as glycolysis/gluconeogenesis, galactose metabolism, or arginine biosynthesis (Fig. 1F). As this observation is the opposite of the WT response to amino acid-rich medium, it suggests a delayed initial adaptation to amino acid-rich medium in *stp2Δ* (Fig. 1D; Fig. S1C). Interestingly, only a few directly amino acid-related pathways were enriched with more abundant metabolites for *stp2Δ* vs. WT in amino acid-rich medium, such as those for AAA, arginine, alanine, aspartate, and glutamate metabolism (Fig. 1F). Also, the intracellular amino acid levels were relatively stable in *stp2Δ* with a trend toward accumulation (Fig. S1F).

In summary, our data indicate that *stp2Δ* cells show a delayed, albeit focused, induction of the global metabolic rearrangements in response to amino acid-rich conditions compared to the WT. This suggests that *C. albicans* activates compensatory metabolic mechanisms to allow for adaptation to limited amino acid availability.

## Data-driven simulation of *C. albicans* metabolism suggests Stp2-dependent and independent metabolic rewiring upon amino acid restriction

To interrogate the possible mechanisms of Stp2-independent compensatory metabolic adaptation in more depth, we opted for a genome-scale metabolic modeling approach to predict how this adaptation might be achieved and to gain insight into interconnected pathway activities.

As detailed above, the *stp2Δ* strain, strongly impaired in amino acid uptake, was able to adapt its metabolism to amino acid restriction. To identify pathway activities involved in this adaptation, we utilized our genome-scale metabolic model (GEM) of *C. albicans* to simulate the metabolic flux in the investigated conditions (35). In brief, we conducted a flux variability analysis (FVA), which predicts the possible range of metabolic flux through all reactions incorporated in the model to maximize biomass production upon a defined diet profile. We proceeded only with the defined media SD and SC for simulations, since CAA composition is naturally varying.

We first performed FVA for WT and *stp2Δ* based on both media compositions of SD and SC. To simulate the uptake deficiency of *stp2Δ*, we limited the uptake rates for all amino acids present in SC as determined previously (Fig. S2A) (30). While the predicted cellular metabolic flux for WT and *stp2Δ* was identical in SD (as the mutation did not perturb the uptake of nutrients present in SD), we observed substantial differences in SC (Fig. S2B). We next integrated our experimentally obtained metabolic profiles for further parameterization of the model and to more precisely reflect the actual metabolic state of the cells. In addition to biomass generation, we constrained the model to generate at least the measured amount of all mapped intracellular metabolites present in the respective conditions. With these adjustments, we repeated the FVA individually for each replicate of the metabolic profiling for the WT and *stp2Δ* in SD and SC. While the integration of experimental data had less impact on our simulations (Fig. S2C), we observed substantial media-dependent differences in feasible flux ranges, particularly present in *stp2Δ* in SC in the data-sensitive predictions (Fig. S2D and E). Those substantially affected reaction fluxes in pathways like AAA, arginine and proline, and purine metabolism. Data-driven metabolic modeling thus showcased so far unknown influences of Stp2 on the fungal response to amino acid limitations.

## Metabolic modeling points to an essential role of the shikimate pathway during amino acid starvation

In accordance with the findings from the metabolic profiling, the PCA of the flux analysis showed clear strain and media-specific separation (Fig. S3A). Unless otherwise noted, we restricted subsequent analysis to all metabolic reactions capable of carrying non-zero flux in at least one of the modeled conditions. Next, using our FVA-predicted reaction flux ranges, we identified four distinct metabolic activity clusters (Fig. 2A). The broadest differences were observed for cluster I, corresponding to different metabolic fluxes in the WT and *stp2Δ* in SC medium.

Subsequently, we identified metabolic reactions with significantly different fluxes in *stp2Δ* vs. WT in amino acid-rich SC medium and calculated the fraction of differential active reactions for each metabolic pathway. In total, we predicted 39 pathways with at least 20% reactions with differing flux ranges in *stp2Δ* vs. WT (Fig. 2B; Fig. S3B). Of note, all pathways except for galactose metabolism identified via the metabolite enrichment analysis also displayed altered metabolic activity in our model predictions, highlighting the predictability of metabolic pathway activity by our data-driven *in silico* approach (Fig. 1F; Fig. S3B). In addition to the enrichment analysis, we found amino acid-related pathways, including histidine, cysteine, and methionine or BCAA metabolism (Fig. 2B).

When investigating only reactions from cluster 1, which comprised the main difference between WT and *stp2Δ* in SC, the most prominently enriched pathway was AAA metabolism, which associated with significantly different flux capabilities for every involved reaction (Fig. 2A, Fig. 2C). Furthermore, we observed substantial changes for fatty acid metabolism, in particular glycerophospholipid metabolism. These simulations reflect pathways with altered activity in the WT upon switching from amino acid-free (SD) to amino acid-rich (SC) medium, which again point to delayed metabolic adaptation in *stp2Δ* (Fig. 1D).

Lastly, we predicted which model-defined metabolic reactions are essential (zero flux is prohibited) for biomass production, given our FVA simulations. While most essential reactions were present in both SD and SC media, we also identified 52 specifically essential reactions for *stp2Δ* in SC and only 10 for the WT (Fig. 2D; Table S2). Interestingly, the *stp2Δ*-specific set contained 10 metabolic reactions assigned to AAA metabolism, for example, the synthesis of 3-deoxy-D-arabinoheptulosonate-7-phosphate (DAHP), which is the entry step to the shikimate pathway (SHKP) required for the synthesis of AAAs, as well as the following steps of this pathway (Fig. 2E; Table S2). This again suggested a prominent role of this pathway in adaptation to *stp2Δ*-induced amino acid starvation. Additionally, we found multiple reactions from biotin, BCAA, serine, and threonine metabolism. In general, we identified an increased number of media-independent and SC-specific metabolic reactions in *stp2Δ* compared to the WT, likely reflecting the activity of pathways necessary to compensate for shortcomings in nutrient uptake (Fig. S3C). Interestingly, when we compared the essential reactions in conditions not providing sufficient amounts of extracellular amino acids—*stp2Δ* in SC and WT in SD—we found that the only essential reactions in both conditions comprised the 10 reactions related to AAA metabolism and three reactions related to glycine, serine, and threonine metabolism (Fig. S3D).

In summary, our data-driven metabolic modeling approach recapitulated and expanded findings of our metabolic profiling in response to the availability of exogenous amino acids. We identified prominent metabolic rearrangements in amino acid-starved *stp2Δ* cells, consisting of different amino acid, lipid, and vitamin-associated pathways. Most importantly, we observed strongly altered flux and predicted essentiality for reactions mapping to the metabolism of AAAs.

## Fungal transcriptomics indicate Stp2-mediated repression of the *ZCF25* MGC

To gain more insights into the molecular mechanisms behind Stp2-mediated metabolic adaptations in *C. albicans* and clarify whether the metabolic rearrangements detailed

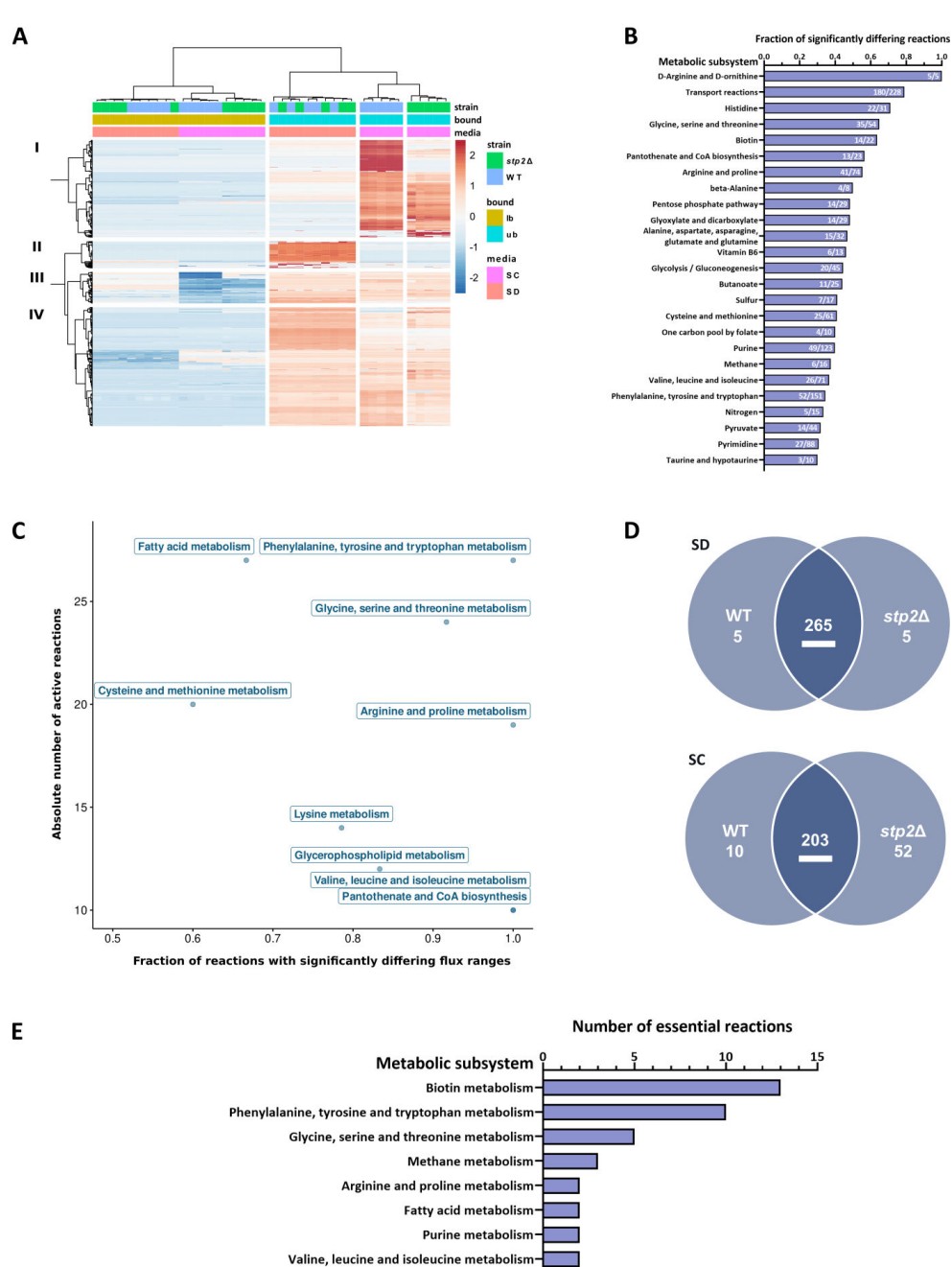

**FIG 2** Predicted metabolic flux in WT and *stp2* Δ in amino acid-free and amino acid-rich medium. (A) Cluster analysis of metabolic reactions (*y*-axis) in SD and SC for WT and *stp2*Δ, carrying flux in at least one condition. Indicated are minimal (lb) and maximal (ub) flux, respectively. (B) Metabolic reactions carrying significantly (*P* < 0.05) different metabolic flux in *stp2*Δ vs. WT in SC as fractions of their respective metabolic subsystems (Top 25). (C) Scatter plot indicating the most enriched metabolic subsystems of reactions (Top 9) carrying metabolic flux from cluster I of panel A. Presented are absolute numbers of active reactions and the fraction of the respective subsystem. Reactions associated with "Transport reactions" were not included in the plot. (D) Comparison of reactions predicted to be essential in WT and *stp2*Δ. (E) Number of essential reactions from the indicated metabolic subsystems specific for *stp2*Δ in SC. Only subsystems with at least two essential reactions are displayed.

above are reflected on the transcriptional level, we next examined the transcriptomes of the WT and *stp2*Δ in SD and SC medium (transcriptome results I).

We found a clear PCA-based separation of expression data between SD and SC medium and for both strains in SC, consistent with the findings of the metabolic

profiling and modeling (Fig. 3A; Fig. S1B and S3A). Consequently, only a small number of differentially expressed genes (DEGs) were detected for stp2Δ vs. WT in SD (62 genes—23 upregulated and 39 downregulated), whereas broad differences were found in SC (1,200 genes—647 upregulated and 553 downregulated) (Fig. 3B; Table S3). Interestingly, following a gene ontology (GO) analysis with the DEGs in stp2Δ vs. WT in SC, we identified primarily general metabolic terms enriched, like organic acid metabolic process or related metabolite transport and mitochondrial function, rather than regulation of specific pathways, with the notable exception of arginine metabolic process and arginine biosynthetic process (Table S3). The term "glutamine family amino acid biosynthetic process," containing genes associated with glutamine, proline, and arginine metabolism, was also enriched (Table S3). Furthermore, three genes associated with arginine metabolism (CAN2, a predicted arginine permease, and ARG1 and ARG3, encoding key enzymes of arginine biosynthesis) were among the set of 37 genes with differential expression in stp2Δ vs. WT for both media, SD and SC (Fig. 3C) (36–39). These findings point to tight Stp2-mediated regulation of arginine synthesis in addition to uptake (Fig. S2A), further strengthened by the observed altered arginine biosynthesis activity in the metabolic profiling and in silico modeling (Fig. 1F and 2B) (30).

Most strikingly, the transcriptional analysis for stp2Δ revealed a significant upregulation of genes from two MGCs, previously linked to the assimilation of hydroxybenzenes, like phenol, catechol, or hydroquinone (Table S3) (31). C. albicans can utilize those metabolites as sole carbon sources by activating the 3-oxoadipate pathway (3OAP), either via the hydroxyhydroquinone (HHQ) or the catechol branch. The genes encoding for these branches are organized within the two MGCs, each of which contains a zinc-finger transcription factor—ZCF10 for the HHQ and ZCF25 for the catechol branch, respectively (Fig. 4A) (31).

In our transcriptional profiling for stp2Δ vs. WT, we found significant upregulation of the entire ZCF25 gene cluster in SC and partially also in SD, indicating a repressive activity of Stp2 on the expression of this cluster (Fig. 4B; Table S3). The ZCF10 gene cluster was only partially induced in stp2Δ vs. WT in SC, suggesting a lesser influence of Stp2 on the regulation of this cluster. Furthermore, the ZCF25 cluster also showed significant upregulation in the WT upon incubation in amino acid-rich medium, implying a role of the 3OAP in the associated metabolic adaptation (Fig. 4B; Table S3). Yet, the metabolites feeding into the 3OAP, including catechol or hydroquinone, are not components of the tested media nor are they commonly found in the niches inhabited by the fungus, suggesting a cellular origin. Since our earlier metabolic modeling simulations predicted for stp2Δ metabolic flux differences in AAA metabolism, compounds structurally related to catechol and hydroquinone, we next investigated potential connections to the 3OAP in C. albicans.

## Sequence similarity analysis predicts a connection between the 3OAP and SHKP in C. albicans

The central and shared element of the de novo synthesis of the three AAAs—phenylalanine, tyrosine, and tryptophan—is the SHKP, which is present in plants and most microorganisms but absent in animals (40). In various organisms, including fungi, this pathway has also been linked to the 3OAP (41–43).

To determine whether the SHKP is also linked to the 3OAP in C. albicans, we first examined the possible connections between both pathways. Thereby, we focused on reactions generating hydroquinone or catechol, as these are expected educts of the two 3OAP branches present in C. albicans (Fig. 4B). Analyzing sequence similarity with BLASTp did not yield a C. albicans protein with quinate dehydrogenase activity (EC 1.1.5.8), which is required for the conversion of quinate to hydroquinone. However, using the catechol-generating 1,2-anthranilate dioxygenase large subunit antA (EC 1.14.12.1) in P. aeruginosa as a query sequence, we identified a potential ortholog encoded by the so far uncharacterized gene C4_00,290C (orf19.5655) (Table S4) (44, 45). Interestingly,

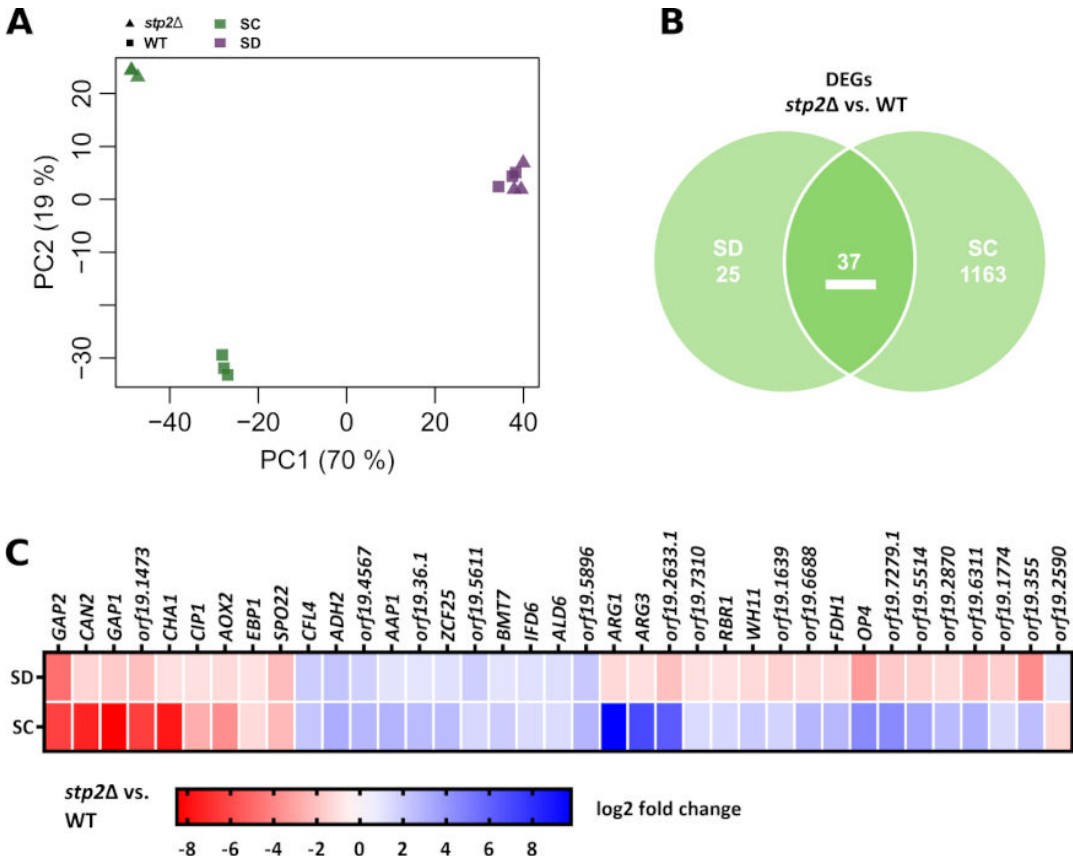

**FIG 3** Transcriptional profiling of WT and *stp2Δ* in amino acid-free and amino acid-rich medium. (A) PCA of WT and *stp2Δ* transcriptomes incubated in SD and SC. (B) DEGs in *stp2Δ* vs. WT in SD and SC with a cutoff of $P < 0.05$ and absolute log2(fold change) ≥ 1. (C) Expression changes of the Stp2 core-regulated genes showing differential expression in SD and SC.

*orf19.5655* was also upregulated in *stp2Δ* vs. WT in SC, similar to the genes of the *ZCF25* cluster (Fig. 4B; Table S3).

Together with our data-driven findings from the metabolic profiling and modeling concerning the SHKP, we inferred the following two hypotheses: (i) the SHKP and 3OAP are connected in *C. albicans*; (ii) amino acid and hydroxybenzene metabolism are further connected on gene regulatory level, with Stp2 acting as a repressor of catechol branch of the 3OAP, encoded by the *ZCF25* gene cluster. Additionally, we aimed to verify the hypothesis that Zcf10 and Zcf25 are regulators of their respective gene clusters and are required for the metabolism of hydroquinone and catechol, respectively (31).

## Zcf25 is the key regulator of catechol metabolism in *C. albicans* and co-regulates 3OAP and SHKP together with Stp2

To verify that Zcf10 and Zcf25 regulate hydroquinone and catechol gene clusters, respectively, we next created deletion strains of *ZCF25*, *ZCF10*, *orf19.5655*, and combinatorial mutants (*zcf10Δ/zcf25Δ* and *zcf10Δ/zcf25Δ/stp2Δ*) and tested their growth behavior in SD, SC, and in media with catechol or hydroquinone as the sole carbon source. Indeed, we found that *zcf25Δ* showed impaired growth on catechol, but not hydroquinone, verifying its role as the regulator of the 3OAP catechol branch (Fig. 5A). Furthermore, no growth defects were observed on SD or SC medium, indicating the dispensability of *ZCF25* in these conditions. For *zcf10Δ* and *orf19.5655Δ*, no growth defects were detected (Fig. 5A). In particular, the absence of a growth defect of *zcf10Δ* on hydroquinone indicates the activity of other regulators on the HHQ branch. Indeed, a slight growth

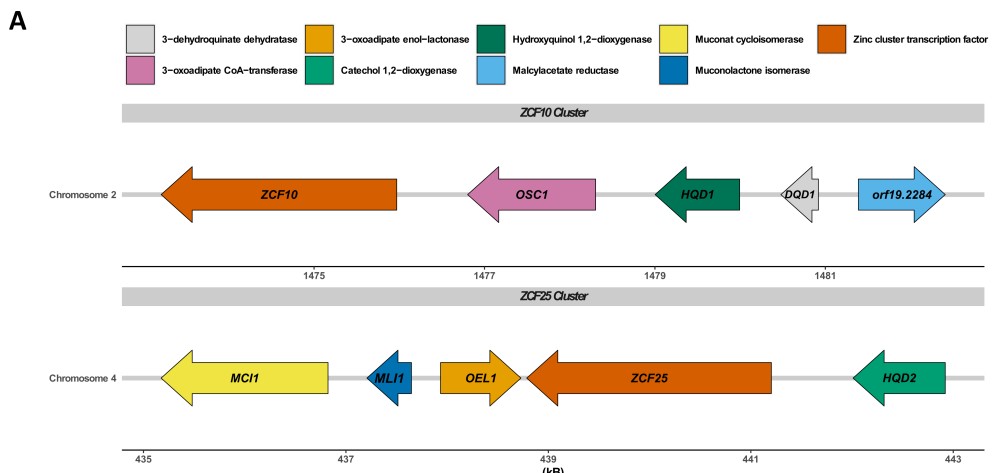

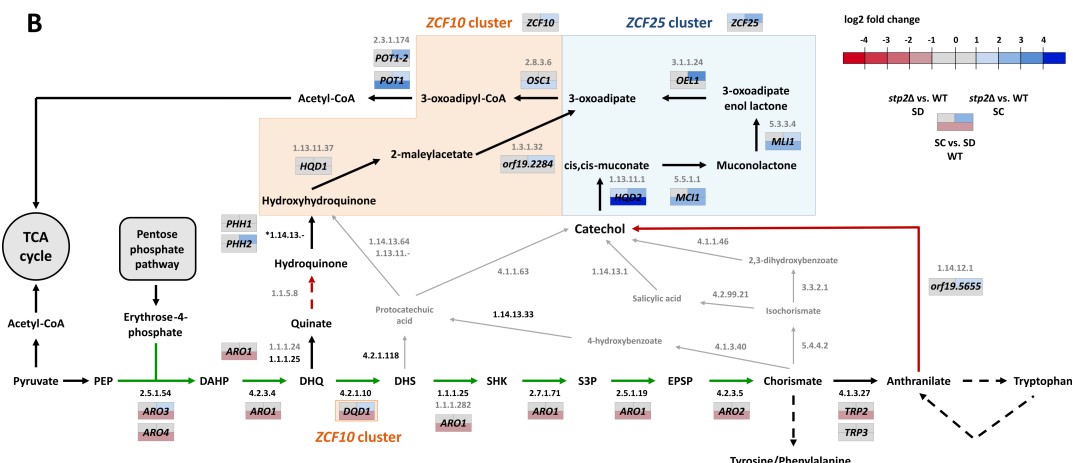

**FIG 4** Proposed connections of 3-oxoadipate and SHKPs in *C. albicans.* (A) Schematic overview of the *ZCF10* and *ZCF25* gene cluster in *C. albicans*, encoding the enzymes of the HHQ- and catechol-branch of the 3OAP with their putative functions. Figure modified after references 30, 31. (B) Overview of the two branches of the 3OAP and the SHK pathway in *C. albicans*. Genes from the *ZCF25* gene cluster and associated reactions are framed in blue, and genes and associated reactions from the *ZCF10* cluster are in orange. Three-part rectangles indicate log2(fold changes) of the respective genes in the indicated comparisons ($P < 0.05$). Metabolic reactions are indicated by arrows, with EC reaction identifiers in black if included in the metabolic model and gray if absent. Green arrows indicate reactions predicted to be specifically essential for *stp2Δ* in SC. Red arrows indicate two possible connections of the SHK and 3OAP in *C. albicans*—to catechol and hydroquinone, while only for the former, a potentially associated gene (*orf19.5655*, encoding a putative 1,2-anthranilate-dioxygenase) was identified via BLAST analysis. Other potential connections of both pathways, likely absent in *C. albicans*, are indicated in gray. Abbreviations: PEP, phosphoenolpyruvate; DAHP, 3-deoxy-D-arabinoheptulosonate-7-phosphate; DHQ, dehydroquinate; DHS, dehydroshikimate; SHK, shikimate; S3P, shikimate-3-phosphate; EPSP, 5-enolpyruvyl-shikimate-3-phosphate.

impairment was noted for *zcf10Δ/zcf25Δ*, pointing to the potential influence of Zcf25 on hydroquinone utilization. Of note, this strain also showed slightly better growth on catechol than the *zcf25Δ* single deletion strain (Fig. 5A).

To investigate the suspected regulatory activity of Zcf25 in catechol utilization and to identify potential other genome-wide influences of this factor and specifically its interplay with Stp2, we performed a second RNA-Seq (transcriptome results II). Toward these aims, we cultivated WT, *stp2Δ*, *zcf25Δ*, and a *stp2Δ/zcf25Δ* in medium containing 10 mM glucose, 1% SC, 5 mM catechol, or 10 mM hydroquinone as the sole carbon source (Fig. S4A). In general, the WT formed distinct clusters for all four tested media, indicating specific transcriptional adaptation (Fig. 5B). Indeed, the WT showed substantial media-specific upregulation and downregulation of genes compared to SD (Fig. S4B). We observed only media-dependent, but not strain-dependent, distinct clusters for SD

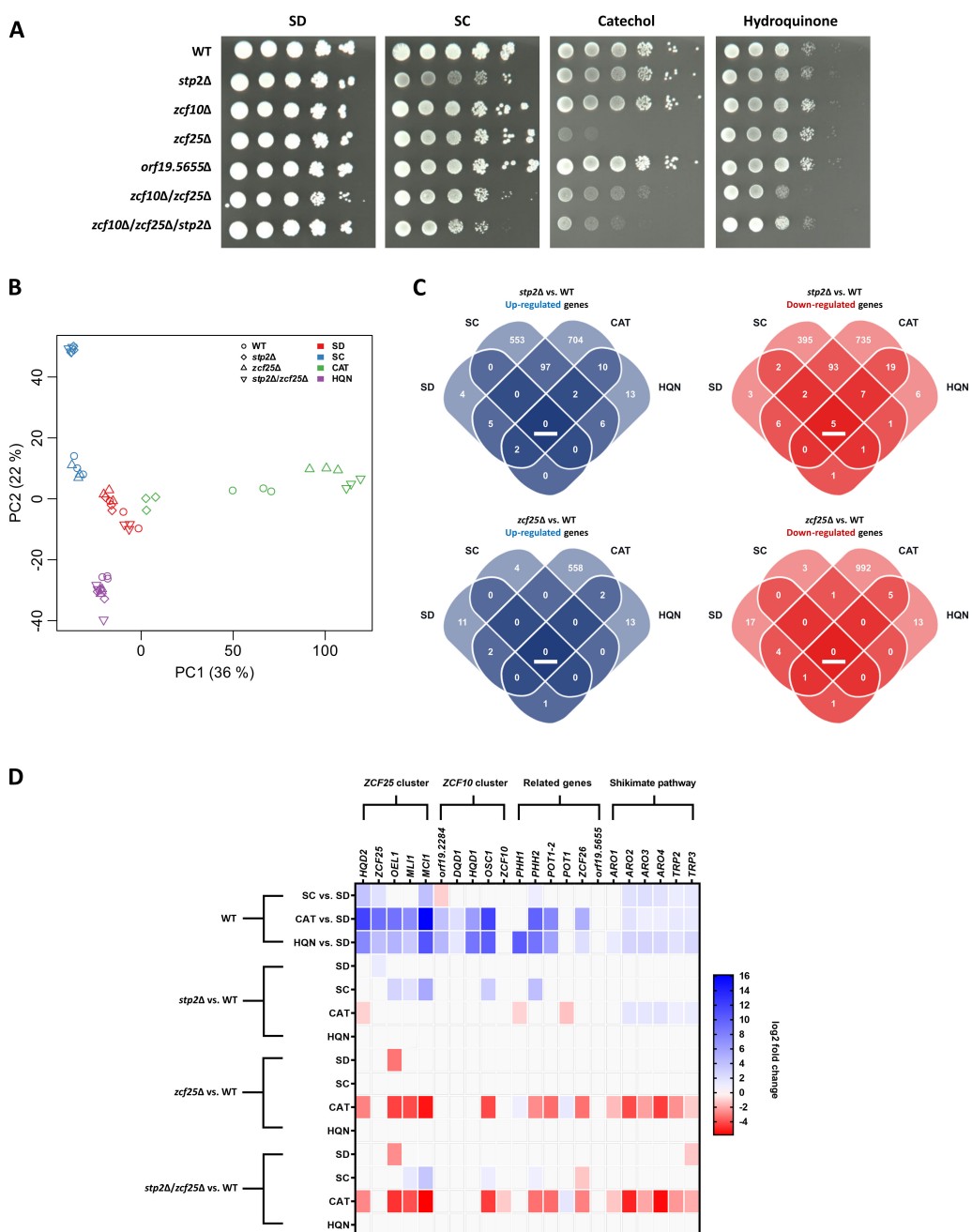

**FIG 5** *Zcf25* is a catechol-specific regulator of the 3OAP and is repressed by Stp2. (A) Serial dilutions of the indicated strains were spotted on agar plates containing the indicated C source (2% glucose—SD, 1% SC, 5 mM catechol, 10 mM hydroquinone) supplemented with YNB and ammonium sulfate. Plates were incubated at 30°C for 3 days. (B) PCA of the transcriptomes from the examined conditions. (C) Significantly upregulated or downregulated genes (*P* > 0.05; absolute log2(fold change)≥1) in *stp2Δ* vs. WT and *zcf25Δ* vs. WT in the indicated media. (D) Expression changes of the 3OAP and SHKP-associated genes in the examined conditions. Changes not meeting the cutoff criteria [*P* > 0.05; absolute log2(fold change) ≥ 1] are indicated in gray.

and hydroquinone in PCA-based analysis, indicating only minor, if any, activity of Stp2 or Zcf25 in these media (Fig. 5B). For catechol, clearly separated clusters were found indicating substantial strain-specific effects (Fig. 5B). Interestingly, *stp2Δ* in catechol clustered close to SD media samples, suggesting that *stp2Δ* gene expression in catechol partly resembles the gene expression associated with amino acid-free SD medium. Nonetheless, WT catechol samples were well separated from SD samples, and Stp2 was

found dispensable for growth on catechol (Fig. 5A and B). Furthermore, *stp2Δ* and *zcf25Δ* in catechol clustered well apart from each other, while *stp2Δ/zcf25Δ* clustered closely to the latter single deletion strain, indicating similar gene expression and a dominant role of Zcf25 activity over Stp2 in catechol. In SC, no separation between the WT and *zcf25Δ* was found, while *stp2Δ* together with *stp2Δ/zcf25Δ* formed one distinct separate cluster, which indicates that Zcf25 activity in amino acid-rich SC medium is minor and likely independent of Stp2-mediated effects (Fig. 5B).

In addition to the expected broad differential gene expression of *stp2Δ* in SC, we found an even greater number of upregulated and downregulated genes in response to catechol (Fig. 5C; Table S5). Notably, the transcriptional response of *stp2Δ* was mostly distinct between catechol and SC, indicating a media-specific Stp2 activity. The *zcf25Δ* deletion strain displayed substantial differential gene expression in response to catechol, further validating the regulatory function of Zcf25 in this condition (Fig. 5C). Interestingly, in the WT, the *ZCF10* and *ZCF25* clusters were both induced in response to hydroquinone or catechol, indicating cross-regulation of the two 3OAP branches (Fig. 5D; Table S5). Yet, Zcf25 is only required for the upregulation of genes from its own cluster as well as *OSC1* and *POT1-2*, which mediate the final steps of 3-oxoadipate degradation to acetyl-CoA (Fig. 4B and 5D).

In response to catechol, only a small set of genes was consistently upregulated or downregulated in all three mutant strains (*stp2Δ*, *zcf25Δ*, and *stp2Δ/zcf25Δ*) compared to the WT (Fig. S4C). In fact, we found that the transcriptome of *stp2Δ/zcf25Δ* was notably different from that of *stp2Δ* in catechol (1,244 upregulated and 1,515 downregulated genes), while it was similar in the other tested conditions, which indicates a dominant role of Zcf25-mediated regulation over Stp2 on gene expression if catechol is the only available carbon source (Fig. S4D). Also, 406 out of 820 genes upregulated in *stp2Δ* in response to catechol were downregulated in *zcf25Δ*, pointing to a catechol-specific antagonistic effect of both transcription factors, with Zcf25 acting as an activator and Stp2 as a repressor (Fig. S4E). Furthermore, the SHKP genes and *TRP2* and *TRP3*, required for anthranilate synthesis, displayed Zcf25-dependent induction in catechol, while in *stp2Δ*, the respective genes showed increased expression (Fig. 5D). Lastly, *orf19.5655* encoding the suspected connecting enzyme of the 3OAP and SHKP, catalyzing the conversion of anthranilate to catechol, showed no differential expression in any of the investigated conditions (Fig. 5D; Table S5). Also, cells exposed to sublethal concentrations of anthranilate showed no upregulation or downregulation of this gene, pointing to a low, yet constant, expression on the gene level (Fig. S4F).

Taken together, our phenotypic and transcriptional analyses strongly suggest Zcf25 as the positive regulator of catechol utilization by *C. albicans* and its own gene cluster. Furthermore, we found evidence of a repressive activity of Stp2 on Zcf25-mediated gene expression in the presence of extracellular amino acids and catechol, as well as a co-regulation of the 3OAP and SHKP by Zcf25 in the latter condition.

## DISCUSSION

Metabolomics and *in silico* approaches successfully contributed to our understanding of the implications that metabolic plasticity has on *C. albicans* virulence traits and commensalism (35, 46–49). In this study, we combined metabolomics, transcriptomics, and *in silico* predictions to detail out fungal adaptation to amino acid availability, critical for fungal virulence, and its interaction with the human host (11, 13). Our analyses focused on the metabolic changes in the fungal response to amino acid starvation using the uptake-deficient strain *stp2Δ*, whose growth is hindered in amino acid-rich medium (23). Our results confirm a prolonged adaptation phase of *stp2Δ* cells followed by re-establishment of metabolic homeostasis in response to amino acid starvation (29, 30). For this, the Stp2-dependent ability to rapidly sense and utilize amino acids, and thus ensure persistence in dynamic host conditions, appears critical (17, 21, 22, 50, 51). However, our most striking findings were the predicted essentiality and increased metabolic flux through the SHKP in *stp2Δ* cells, accompanied by a strong upregulation of

genes associated with the 3OAP—proposing a functional and regulatory connection of both metabolic pathways.

The 3OAP consists of distinct branches and allows the detoxification and assimilation of different hydroxybenzoates and hydroxybenzenes, like phenol, present in the environment (31, 52). The final product, acetyl-CoA, can then be either used for the synthesis of new molecules or funneled into the TCA cycle for energy generation. The pathway is commonly found in environmental bacteria and fungi, which use it, for example, for lignin degradation (42, 53–56). It is also frequently present in the species of the CUG clade, although their genetic repertoire and capacity to assimilate the diverse educts differ significantly (31). Several studies addressed the mechanisms of 3OAP regulation in *C. parapsilosis*, which lacks the catechol branch in comparison to *C. albicans* and instead employs an expanded version of the HHQ branch and the related gentisate pathway to utilize hydroxybenzoates (57–60). Yet, while the presence of this pathway appears reasonable in a fungus like *C. parapsilosis*, which is not as strictly associated with its human host as *C. albicans*, its biological role for those fungi remains an open question (61).

The sole fact that *C. albicans* kept both 3OAP MGCs present in the presumed ancestor of the serinales, while species like *C. parapsilosis* lost the catechol-branch, argues for an important function within its opportunistic lifestyle (31, 62). So far, only a few studies have investigated *C. albicans* *ZCF25* activity, showing mixed results. While a heterozygous deletion led to decreased fitness, a homozygous deletion resulted in increased or unaltered fitness based on the readout (63, 64). Another study found increased *C. albicans* resistance against fluconazole when *ZCF25* possessed a gain-of-function mutation (65). Lastly, *ZCF25* possesses at least two paralogs, *ZCF15* and *ZCF26*, with the latter residing directly upstream of the *ZCF25* cluster (66). Both reportedly impact cellular metabolism, fungal stress responses, and biofilm formation (66, 67).

Intriguingly, the genes associated with the 3OAP in *C. albicans* and other species from the CUG clade are encoded in MGCs (31). While MGCs are widespread in most fungal phyla (especially those with broad secondary metabolism), *Candida* species contain only very few (68, 69). Among them, *C. albicans* possesses two MGCs that encode either the HHQ or the catechol branch of the 3OAP (31). Since both clusters also carry a zinc-cluster transcription factor, it strongly suggests a feedforward regulation. Indeed, we found a strong dependence of *C. albicans* on Zcf25 for growth on catechol and induction of its respective cluster genes, as well as *OSC1* and *POT1-2* from the *ZFC10* cluster, which are both required for catechol degradation to acetyl-CoA. These findings verified Zcf25 as a condition-specific activator of its own gene cluster. Notably, Zcf10 was expendable for growth on hydroquinone, a finding that aligns with a previous report showing that the deletion of the *C. parapsilosis* *ZCF10* ortholog, *OTF1*, barely affects growth on hydroquinone or the expression of its respective gene cluster (59). However, the *ZCF25* gene cluster also displayed strong induction in response to hydroquinone in both WT and *zcf25Δ*, indicating the activity of further regulators on this cluster. Interestingly, the aforementioned Zcf25 paralogs, Zcf16 and Zcf26, were shown to bind upstream of the *ZCF25* and *HQD2*, encoding the catechol 1,2-dioxygenase, open reading frames (67). Thus, an influence of both on Zcf25 regulation and 3OAP induction appears likely. Furthermore, we also observed notable induction of the putative phenol hydroxylases *PHH1* (CR_07,940W) and *PHH2* (CR_07,820W) in response to hydroquinone. While we confirmed that Zcf10 is actually expendable for growth on hydroquinone, we observed a partially Zcf25-dependent induction of *PHH2* in response to catechol, indicating Zcf25 regulatory influence on phenol catabolism. In this study, we also observed a clear regulatory influence of Stp2 on the 3OAP, which represses the *ZCF25* cluster in amino acid-rich and partially in amino acid-free medium. Interestingly, this activity was only observed in SC medium and not in response to catechol. Yet, a high number of genes also showed Stp2-dependent expression changes in response to catechol, with those genes being largely distinct from the ones affected in amino acid-rich SC medium, pointing to condition-specific activity of Stp2. These potential novel regulatory activities

of Stp2 require future experimental validation, exemplary in the direction of extracellular amino acids in general or specific sets, like AAAs, influence 3OAP activity.

The origin of the educts feeding into the catechol- and HHQ-branch in *C. albicans* is unclear. Since an extracellular origin appears unlikely, due to the close association of the fungus with its human host, an intracellular origin seems plausible. Based on the outlined results, we suspected a direct metabolic link between the SHKP and 3OAP in *C. albicans*, as it is also reported for other organisms. *Aspergillus nidulans*, for example, assimilates carbon from exogenous shikimate and quinate via protocatechuate and the 3OAP, whereas *Pseudomonas aeruginosa* likely uses anthranilate for energy production via degradation to catechol (41–43). Furthermore, links between hydroxybenzene and AAA metabolism are established and used in various organisms in the context of bioengineering, thereby illustrating potential connections (70–72). A potential link in *C. albicans* could be via the putative 1,2-anthranilate dioxygenase encoded by *orf19.5655*. However, this remains to be verified, since *orf19.5655Δ* did not show a prominent phenotype in the conditions tested, nor gene expression differences here or in most publicly available data sets (25, 49, 73). Although *orf19.5655* is apparently not regulated by transcription, it also needs to be investigated whether its activity is mediated by post-transcriptional regulatory mechanisms.

However, we previously found an upregulation of *orf19.5655* during WT, and even more pronounced in *stp2Δ*, biofilm formation, suggesting a role in adaptation to limited nutrient availability (30). Similarly, we found evidence that the proposed link between SHKP and 3OAP is involved in the metabolic adaptation of *C. albicans* to changes in amino acid availability. Exemplary, the catabolism of excess tryptophan would be possible through the degradation to catechol via anthranilate. This notion is supported by the partial upregulation of the *ZCF25* cluster in amino acid-rich medium (Table S3). Given that tryptophan uptake is also strictly Stp2-dependent, a concurrent repression of the *ZCF25* cluster by Stp2 would allow fine-tuning of potential simultaneous uptake and degradation of this costly amino acid (30, 74). Furthermore, a study showing that *C. albicans* uses a type II 3-dehydroquinate dehydratase within the SHKP encoded by *DQD1* in the *ZCF10* gene cluster instead of Aro1, the SHKP multifunctional main enzyme, supports the hypothesized connection between both pathways (75). Lastly, we observed catechol-specific downregulation of SHKP genes in *zcf25Δ*, arguing for another layer of regulatory connections between SHKP and 3OAP.

In conclusion, a number of questions remain unanswered, for example—Is the catechol feeding the 3OAP in *C. albicans* of endogenous or exogenous origin? If exogenous, where does it come from in the host, and how is it sensed and transported into the fungus? Which enzyme(s) links the 3OAP and SHKP? How are central regulatory circuits, like nitrogen or carbon catabolite repression, affecting these potential connections and the 3OAP in general (9)? Answering these questions will further elucidate the physiological role of the 3OAP in the opportunistic lifestyle of *C. albicans*, ultimately contributing to our growing understanding of the underlying, specific metabolic traits allowing the close association with the human host and thus helping us to follow new avenues for therapeutic control of this pathogen.

## MATERIALS AND METHODS

### Cultivation of *C. albicans*

All chemicals and media components used in this study were purchased from Sigma Aldrich if not stated explicitly otherwise. Routinely, *C. albicans* strains were cultivated in YPD (1% yeast extract, 2% peptone, 2% glucose, solidified with 2% agar for plates). Overnight cultures were prepared in liquid YPD or SD (0.17% yeast nitrogen base without ammonium sulfate and amino acids, 0.5% ammonium sulfate, 2% glucose, pH 4.5) medium at 30°C with 180 rpm shaking, washed twice with, and resuspended in sterile $dH_2O$ prior to experimental use unless otherwise noted.

## Strain construction

All deletion mutants used in this study were created in the *C. albicans* wild-type strain SC5314 according to the *SAT1-FLP* method (76). Strains and utilized gene-specific oligonucleotides are listed in Table S6.

## Growth assays on solid medium

*C. albicans* cells from SD ON cultures were resuspended in sterile $dH_20$ and t10-fold dilution series were prepared starting at OD 1. 5 µL per strain, and the dilution was spotted on solid test media and incubated for 3 days at 30°C, and images wer taken with a ProtoCol2 colony counter (Synbiosis, UK). The test media were prepared as follows: 0.17% yeast nitrogen base without ammonium sulfate and amino acids, 0.5% ammonium sulfate, 2% purified Bacto agar (BD, New Jersey, USA), pH 6. To this base medium, one of the following indicated carbon sources was added: 2% glucose, 1% SC (US Biological D9515, purchased via Biomol, Germany), 10 mM hydroquinone, or 5 mM catechol. All media recipes are also listed in Table S6.

## Transcriptional profiling

### Isolation of RNA and qRT-PCR

Total RNA extractions in this study were done via phenol-chloroform extraction as recently described (17). RNA quality and quantity were assessed using 2100 Bioanalyzer (Agilent Technologies, California, USA) and Nanodrop (Thermo Fisher Scientific, Massachusetts, USA) measurements. Quantitative RT-PCR was performed as recently detailed, to determine expression levels of *orf19.5655* according to the Δct method using *MED15* as the internal reference (17, 77). Gene-specific primers are listed in Table S5. ON cultures of the indicated strains were prepared in SD and re-cultivated in SD until early log phase was reached (OD 1). Test media—SD and SC (0.17% yeast nitrogen base without ammonium sulfate and amino acids, 0.5% ammonium sulfate, 1% SC, pH 4.5) with or without 2 mM anthranilate—were inoculated at OD 0.3 with the indicated strains and incubated for 1 h.

### RNA sequencing and bioinformatic analysis

To examine transcriptional changes in the WT and *stp2Δ* in response to amino acids, the following setup was used. Cells from YPD ON cultures were re-inoculated in YPD and incubated until early log phase was reached (OD 1). 30 mL of test media was inoculated at an OD of 0.3 and incubated for 60 min at 37°C. Test media were prepared as follows: 0.17% yeast nitrogen base without ammonium sulfate and amino acids, 0.5% ammonium sulfate, pH 4.5 with 1% SC, 1% casamino acids, or 2% glucose as the sole carbon source. To examine transcriptional changes of the WT, *stp2Δ*, *zcf25Δ*, and *stp2Δ/ zcf25Δ* in response to hydroxybenzenes, the following setup was used. Cells from SD ON cultures were re-inoculated in SD and incubated until early log phase was reached (OD 1). The test medium was subsequently inoculated at an OD of 0.3 and incubated for 4 h, and cells were collected for the isolation of RNA. Test media were prepared as follows: 0.17% yeast nitrogen base without ammonium sulfate and amino acids, 0.5% ammonium sulfate, pH 4.5 with 10 mM glucose, 5 mM catechol, 10 mM hydroquinone, or 1% SC as the sole carbon source. Culture conditions are visible at a glance in Table S6. RNA sequencing was performed by GATC (Konstanz, Germany), Genewiz (Leipzig, Germany), and Novogene (Cambridge, UK). For all samples, a minimum of 10 million reads of 150 bp length, paired ends, were sequenced. Bioinformatic analyses were carried out as previously described (49).

## GO and Venn analysis

Gene sets showing significant differences in analyzed contrasts ($P < 0.05$ and absolute log2(fold change)≥1) were analyzed for enriched GO terms (biological process and cellular function) with the GO term finder of the Candida genome Database using standard settings (36). Redundancy of obtained GO terms was reduced via REVIGO with a dispensability cutoff of 0.7 (78). Venn analyses were performed using the online tool https://bioinformatics.psb.ugent.be/webtools/Venn/. Results were visualized with GraphPad Prism 9.4.

## Metabolic profiling

*C. albicans* strains for metabolic profiling were incubated in the same setup used to determine the transcriptional changes of the WT and *stp2Δ* in response to amino acids, with the sole difference that cells were incubated for 90 min in the test medium. Culture condition details are described in Table S6. Metabolite extraction and measurement were carried out by Metabolon, Inc. (Morrisville, NC, USA). Processing of metabolome data was performed as recently described (30, 49). For downstream enrichment analyses, only metabolites possessing an identifier for the human metabolome database (HMDB) (429 out of 586 metabolites) were used (79). KEGG pathway enrichment analyses were performed using MetaboAnalyst (34). Results were visualized with GraphPad Prism 9.4.

## Metabolic modeling

*In silico* metabolic analyses were carried out using the recently published *C. albicans* genome-scale metabolic model (35). The model was parameterized for specific media compounds exemplifying SD and SC media. In brief, only the influx of media-associated metabolites was allowed and unconstrained. In the case of SC media, uptake of glucose was prohibited by setting the respective exchange reaction influx to zero. Uptake rates of amino acids were further parameterized by available mass spectrometry (MS) measurements (30). Next, metabolite concentrations as measured with our metabolome data were integrated into our *in silico* model. A substantial portion of all detected metabolites could be mapped to metabolites defined in our model (252 out of 586, 43%), thus providing a sufficient base for data-driven modeling (Table S2). If no exchange reactions were available for a secreted metabolite, an artificial exchange reaction was added to the model to allow mimicking its secretion in flux simulations. Unless metabolites were part of the influx media composition, secretion rates were adapted from the measured concentrations. Toward this aim, we carried out a FVA to quantify feasible secretion fluxes of measured metabolite concentrations with our model (80). In cases where media uptake already constrained feasible secretion rates, maximum secretion rates were set to the FVA identified values, otherwise to the measured concentration values per replicate. Depending on the media and replicate data for WT or *stp2Δ* mutants, this resulted in replicate-specific models with different feasible influx (depending on media and MS data) and outflux (according to measured concentration data and FVA-derived feasible outflux values) of metabolites.

For each model, we again carried out FVA to quantify feasible internal reaction flux ranges for each defined metabolic reaction per replicate. All FVA simulations were carried out ensuring at least 90% flux of the simulated optimal flux value of the biomass objective function. Moreover, at most an additional 10% of the overall summed-up flux values were allowed to circumvent unrealistic high overall fluxes and thus to constrain the feasible solution space toward optimal and near-optimal usage of fluxes. Simulations were carried out using Python (v3.6.12) and the metabolic modeling toolbox cobrapy (v0.20.0) (81). Gurobi (v9.1.1, https://www.gurobi.com) was used for solving integer linear optimization problems (biomass optimization and FVA). Further data analyses were carried out using the R programming language (v4.2.1).

## Examination of potential connections between AAA and hydroxybenzene metabolism

To determine potential connections, publicly available data were utilized—the proposed pathway (Fig. 4) was created by adapting several KEGG pathways (IDs 00400, 00362, 01220, and 00380) and references 31, 41, 72, 75.

### ACKNOWLEDGMENTS

We thank all members of the group "Host Fungal Interfaces" and the Jena School for Microbial Communication (JSMC) for support of this work and insightful discussions. We further thank Daniel Rosenberger, Mohamed Balboul, and Mahmoud Elbahnasy for excellent technical assistance and Anna Möslinger for proofreading of the manuscript.

This work was supported by the German Ministry for Education and Science in the program Unternehmen Region (BMBF 03Z22JN11) awarded to S.V. This work was supported by the Deutsche Forschungsgemeinschaft (DFG; www.dfg.de) CRC/Transregio 124 "Pathogenic fungi and their human host: Networks of interaction" (DFG project number 210879364, subproject INF to S.S., and G.P., subproject C1 to B.H. and subproject C2 to S.V. and B.H.). T.B.S. was funded by the Deutsche Forschungsgemeinschaft (DFG; German Research Foundation) under Germany's Excellence Strategy—EXC 2051— Project-ID 390713860.

### AUTHOR AFFILIATIONS

[1]Septomics Research Center, Friedrich Schiller University and Leibniz Institute for Natural Product Research and Infection Biology–Hans Knöll Institute, Jena, Germany
[2]Department of Microbiome Dynamics, Leibniz Institute for Natural Product Research and Infection Biology–Hans Knöll Institute, Jena, Germany
[3]BioControl Jena, Jena, Germany
[4]Institut Pasteur, Université Paris Cité, INRAE USC2019, Unité Biologie et Pathogénicité Fongiques, Paris, France
[5]Department Microbial Pathogenicity Mechanisms, Leibniz Institute for Natural Product Research and Infection BiologyHans Knöll Institute, Jena, Germany
[6]Cluster of Excellence Balance of the Microverse, Friedrich-Schiller-University Jena, Jena, Germany
[7]Faculty of Biological Sciences, Friedrich Schiller University Jena, Jena, Germany
[8]Faculty of Medicine, Friedrich Schiller University Jena, Jena, Germany
[9]University of California San Francisco, San Francisco, California, USA

### AUTHOR ORCIDs

Enrico Garbe http://orcid.org/0009-0007-3435-6637
Sascha Schäuble http://orcid.org/0000-0003-3862-6546
Bernhard Hube https://orcid.org/0000-0002-6028-0425

### FUNDING

| Funder | Grant(s) | Author(s) |
| --- | --- | --- |
| Bundesministerium für Bildung und Forschung | 03Z22JN11 | Enrico Garbe |
| | | Bettina Böttcher |
| | | Robert Jesse |
| | | Slavena Vylkova |
| Deutsche Forschungsgemeinschaft | 210879364 | Sascha Schäuble |
| | | Bernhard Hube |
| | | Gianni Panagiotou |
| | | Slavena Vylkova |

| Funder | Grant(s) | Author(s) |
|--------|----------|-----------|
| Deutsche Forschungsgemeinschaft | 390713860 | Tim Bastian Schille |

## AUTHOR CONTRIBUTIONS

Enrico Garbe, Conceptualization, Data curation, Formal analysis, Investigation, Methodology, Validation, Visualization, Writing – original draft, Writing – review and editing | Sascha Schäuble, Conceptualization, Data curation, Formal analysis, Funding acquisition, Investigation, Methodology, Validation, Visualization, Writing – review and editing | Bettina Böttcher, Formal analysis, Investigation, Methodology, Writing – review and editing | Robert Jesse, Investigation, Visualization, Writing – review and editing | Dominik Driesch, Formal analysis, Visualization, Writing – review and editing | Lasse van Wilijck, Formal analysis, Investigation, Methodology, Writing – review and editing | Osama Elshafee, Investigation, Writing – review and editing | Tim Bastian Schille, Investigation, Writing – review and editing | Bernhard Hube, Funding acquisition, supervision, Writing – review and editing | Gianni Panagiotou, Funding acquisition, supervision, Writing – review and editing | Slavena Vylkova, Conceptualization, Formal analysis, Funding acquisition, supervision, Writing – review and editing

## DATA AVAILABILITY

Transcriptional raw data are available at GEO (accession nos. GSE277530 and GSE277771).

## ADDITIONAL FILES

The following material is available online.

### Supplemental Material

**Supplemental figures (mSystems00226-25-s0001.docx).** Figures S1 to S4.
**Table S1 (mSystems00226-25-s0002.xlsx).** Metabolome results.
**Table S2 (mSystems00226-25-s0003.xlsx).** Modeling results.
**Table S3 (mSystems00226-25-s0004.xlsx).** Transcriptomics results I: amino acids.
**Table S4 (mSystems00226-25-s0005.xlsx).** BLAST results.
**Table S5 (mSystems00226-25-s0006.xlsx).** Transcriptomics results II: hydroxybenzenes.
**Table S6 (mSystems00226-25-s0007.xlsx).** Strains, oligonucleotides, and media used in the study.

### Open Peer Review

**PEER REVIEW HISTORY (review-history.pdf).** An accounting of the reviewer comments and feedback.

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
