## [Reviewer comments · mSystems]

A multi-omics analysis unveils functional and regulatory links between hydroxybenzene and aromatic amino acid metabolism in *Candida albicans*

Enrico Garbe, Sascha Schäuble, Bettina Böttcher, Robert Jesse, Dominik Driesch, Lasse van Wilijk, Osama Elshafee, Tim Schille, Bernhard Hube, Gianni Panagiotou, and Slavena Vylkova

Corresponding Author(s): Enrico Garbe, Leibniz-Institute for Natural Product Research and Infection Biology- Hans Knoell Institute

Review Timeline:

Submission Date:	February 14, 2025
Editorial Decision:	May 16, 2025
Revision Received:	August 12, 2025
Accepted:	September 16, 2025

Editor: Benjamin Wolfe

Reviewer(s): Disclosure of reviewer identity is with reference to reviewer comments included in decision letter(s). The following individuals involved in review of your submission have agreed to reveal their identity: Jozef Nosek (Reviewer #1)

Transaction Report:

DOI: <https://doi.org/10.1128/msystems.00226-25>

Re: mSystems00226-25 (**A multi-omics analysis unveils functional and regulatory links between hydroxybenzene and aromatic amino acid metabolism in *Candida albicans***)

Dear Dr. Enrico Garbe:

Thank you for the privilege of reviewing your work. Two experts in the field have assessed your manuscript and have suggestions for how you can improve clarity and accuracy of your manuscript. Below you will find their comments and instructions from the mSystems editorial office.

Revision Guidelines

Sincerely,
Benjamin Wolfe
Editor
mSystems

Reviewer #1 (Comments for the Author):

By means of metabolic and transcriptional profiling of human pathogenic yeast *Candida albicans*, Garbe et al. uncovered and characterized a link between the metabolism of aromatic amino acids and the 3-oxoadipate pathway (3-OAP) involved in the catabolism of hydroxybenzenes. They also investigated roles of the transcription factors Stp2, Zcf25, and Zcf10 and a gene coding for a putative anthranilate 1,2-dioxygenase C4_00290C. The genes for Zcf25 and Zcf10 are present in the metabolic gene clusters (MGCs) coding for the 3-OAP enzymes and the authors demonstrated that Zcf25 controls the catechol branch of

the 3-OAP as well as the shikimate pathway. The manuscript is well written, and the results appear sound and well justified.

Minor comments:

1. Lines 346-347: The authors propose a hypothesis that Zcf10 and Zcf25 are involved in the transcriptional control of the 3-OAP MGCs. However, the roles of both transcription factors in the 3-OAP regulation have been proposed about 10 years ago by Gerecova et al. 2015.

2. I would suggest mentioning also the changes in expression of PHH1 and PHH2 genes encoding phenol hydroxylases that result from the knock-outs of ZFC10 and ZCF25 (e.g. in the paragraph on lines 452-469).

3. *C. albicans* assimilates phenol via the catechol branch of the 3-OAP. Although I do not insist on this, it would be great to add to Figure 5 also a panel with the strains utilizing phenol as a sole carbon source.

4. The authors could mention that similar to *C. albicans*, a knock-out of the OTF1 gene in *C. parapsilosis* (an ortholog of ZCF10) also exhibits only mild changes on the transcriptional activation of the corresponding MGC (Cillingova et al. 2022).

5. The expression of genes in the 3-OAP MGCs is repressed in glucose containing media (presumably mediated via Mig1/2). Could the authors clarify whether the presence of aromatic amino acids in cultivation media represses the genes in the 3-OAP MGCs?

6. Line 443: The sentence "... *C. albicans* not only kept HHQ- but also gained the catechol branch of the 3OAP ..." is a bit misleading, as comparative studies (e.g. on *C. subhashii* by Mixao et al. 2021) indicate that a presumed ancestor of the CUG clade possessed both 3-OAP MGCs. Hence, *C. albicans* rather kept both ancestral clusters, while some other species (e.g. *C. parapsilosis*) lost this branch of 3-OAP.

Reviewer #2 (Comments for the Author):

This work by Garbe et al. focuses on the utilization of metabolomics and transcriptomics approach (lumped together as "multi-omics") to unravel potential mechanisms by which the human fungal pathogen *Candida albicans* is able to adapt in medium lacking extracellular amino acids (exAA), enabling the restoration of metabolic balance in the cell. The authors used established analyses pipelines to identify DEGs and altered metabolites in cells grown under specific conditions. Central to this work is the transcription factor Stp2, which has been shown by previous studies, including that of the authors, to mediate the assimilation of extracellular amino acids through the derepression of a specific subsets of amino acid transporters. The work nicely captured the outcome expected of stp2 like its capacity to remain adaptable albeit in a very a delayed manner in AA-rich conditions. The authors found that the stp2 KO in exAA-lacking condition, is characterized by the upregulation of shikimate pathway (aromatic AA biosynthesis) and catabolism of hydroxybenzene, the latter inferred from the upregulation of specific genes clusters linked to 3-oxoadipate (3-OAP) pathway. The 3-OAP is suggested to be controlled by another transcription factor Zcf25, which the authors suggest to be negatively regulated by Stp2.

The work provides sufficient overview and presentation of key differences among strains grown under specific conditions. However, as this work focuses on amino acid metabolic processes and all -omics interpretations were drawn from cells grown under specific conditions, it is important that the specific culture/pre-culture conditions are described and substantiated. In addition, integrating two transcriptomics and metabolomics approaches, need to meet this consistency element to make the data useful in obtaining a comprehensive understanding of biological processes.

Comments:

1. To consider the shifting of cells from complex medium like YPD to SD as an AA-rich to AA-poor adaptation (discussed in line 419-421) is rather a misnomer as ammonium, which is present in all test media, is immediately sequestered by the central nitrogen metabolism to generate key amino acids like glutamine/glutamate, which can then be converted to a plethora of other amino acids. This is nicely reflected in metabolomics data in Fig. S1F. Adding amino acids (1% SC) to the base medium would by-pass this dependency on central nitrogen metabolism, allowing other amino acids to be sequestered by other pathways. If any, the switch is a reflection of the cells need to convert AA to glucose, and not because of AA-limitation/starvation. The use of other non-preferred nitrogen source like allantoin/purines in the base medium, would be more ideal in this regard since they are not immediately converted to key amino acids.

2. All throughout the text, the authors mentioned the use of "SC" as a glucose-free, amino acid-rich condition where Stp2 processing is induced (from line 124). I am unable to trace the specific description of this medium in the text other than the mention of the supplier (Biomol; line 530). Can the authors please clarify this? This is important because not all amino acids are capable of inducing Stp2 processing and understanding the proportion of available amino acids would provide an indication as to how these fluxes would operate. The authors also suggested that under the "SC" condition, the Stp2 is fully required (line 125).

However, the drop plate shown in Fig. 5A, does not fully reflect this as the *stp2* had appreciable growth. Any comments on this? Have the authors checked the phenotypes in liquid cultures as well?

3. It would benefit the readers to restructure the methods section particularly that concerns the transcriptional profiling/RNASeq (e.g., line 533-560) as there are several conditions/experiments described. Can the authors confirm that the preculture conditions (YPD) prior to media switch for RNASeq and metabolomics for WT/*stp2* are the same? Any specific reasons as to why in some cases the authors switched to SD (2% glucose) as pre-culture condition? The biological insights on AAA/HB metabolism were drawn from the first metabolic profiling using YPD as pre-culture.

4. In addition, any reasons as to why the glucose condition was changed from 2% to 10 mM in catechol RNASeq experiment (line 555-557)? This is essentially the focus of Fig. 5. Is this correct? Does the "SD" used as a comparator to assess in gene clustering have 10 mM or 2% glucose? It is well-known that glucose levels has a profound influence on metabolic activity and downstream processes, and I wonder whether the key suggestions like *stp2* gene expression in catechol resembling that of *stp2* in SD medium is still valid?

5. The idea that *Stp2* represses *Zcf25* is interesting. However, caution is needed in over interpreting the data in Fig. 5D as *Stp2* regulation occurs at different level (presence of inducing amino acids, cytosolic interactors, etc.). In addition, only a subset of genes are affected in the *stp2* strain relative to *zcf25* strain. Unless more data is presented that dissect this relationship, it suggested that the authors tone this down.

Garbe et al., **A multi-omics analysis unveils functional and regulatory links between hydroxybenzene and aromatic amino acid metabolism in *Candida albicans***

This work by Garbe et al. focuses on the utilization of metabolomics and transcriptomics approach (lumped together as “multi-omics”) to unravel potential mechanisms by which the human fungal pathogen *Candida albicans* is able to adapt in medium lacking extracellular amino acids (exAA), enabling the restoration of metabolic balance in the cell. The authors used established analyses pipelines to identify DEGs and altered metabolites in cells grown under specific conditions. Central to this work is the transcription factor Stp2, which has been shown by previous studies, including that of the authors, to mediate the assimilation of extracellular amino acids through the derepression of a specific subsets of amino acid transporters. The work nicely captured the outcome expected of stp2 like its capacity to remain adaptable albeit in a very a delayed manner in AA-rich conditions. The authors found that the stp2 KO in exAA-lacking condition, is characterized by the upregulation of shikimate pathway (aromatic AA biosynthesis) and catabolism of hydroxybenzene, the latter inferred from the upregulation of specific genes clusters linked to 3-oxoadipate (3-OAP) pathway. The 3-OAP is suggested to be controlled by another transcription factor Zcf25, which the authors suggest to be negatively regulated by Stp2.

The work provides sufficient overview and presentation of key differences among strains grown under specific conditions. However, as this work focuses on amino acid metabolic processes and all -omics interpretations were drawn from cells grown under specific conditions, it is important that the specific culture/pre-culture conditions are described and substantiated. In addition, integrating two transcriptomics and metabolomics approaches, need to meet this consistency element to make the data useful in obtaining a comprehensive understanding of biological processes.

Comments:

1. To consider the shifting of cells from complex medium like YPD to SD as an AA-rich to AA-poor adaptation (discussed in line 419-421) is rather a misnomer as ammonium, which is present in all test media, is immediately sequestered by the central nitrogen metabolism to generate key amino acids like glutamine/glutamate, which can then be converted to a plethora of other amino acids. This is nicely reflected in metabolomics data in Fig. S1F. Adding amino acids (1% SC) to the base medium would by-pass this dependency on central nitrogen metabolism, allowing other amino acids to be sequestered by other pathways. If any, the switch is a reflection of the cells need to convert AA to glucose, and not because of AA-limitation/starvation. The use of other non-preferred nitrogen source like allantoin/purines in the base medium, would be more ideal in this regard since they are not immediately converted to key amino acids.
2. All throughout the text, the authors mentioned the use of “SC” as a glucose-free, amino acid-rich condition where Stp2 processing is induced (from line 124). I am unable to trace the specific description of this medium in the text other than the mention of the supplier (Biomol; line 530). Can the authors please clarify this? This is important because not all amino acids are capable of inducing Stp2 processing and understanding the proportion of available amino acids would provide an indication as to how these fluxes would operate. The authors also suggested that under the “SC” condition, the Stp2 is fully required (line 125). However, the drop plate shown in Fig. 5A, does not fully reflect this as the stp2 had appreciable growth. Any comments on this? Have the authors checked the phenotypes in liquid cultures as well?
3. It would benefit the readers to restructure the methods section particularly that concerns the transcriptional profiling/RNASeq (e.g., line 533-560) as there are several conditions/experiments described. Can the authors confirm that the preculture conditions

(YPD) prior to media switch for RNASeq and metabolomics for WT/stp2 are the same? Any specific reasons as to why in some cases the authors switched to SD (2% glucose) as pre-culture condition? The biological insights on AAA/HB metabolism were drawn from the first metabolic profiling using YPD as pre-culture.

4. In addition, any reasons as to why the glucose condition was changed from 2% to 10 mM in catechol RNASeq experiment (line 555-557)? This is essentially the focus of Fig. 5. Is this correct? Does the "SD" used as a comparator to assess in gene clustering have 10 mM or 2% glucose? It is well-known that glucose levels have a profound influence on metabolic activity and downstream processes, and I wonder whether the key suggestions like *stp2* gene expression in catechol resembling that of *stp2* in SD medium is still valid?
5. The idea that *Stp2* represses *Zcf25* is interesting. However, caution is needed in over interpreting the data in Fig. 5D as *Stp2* regulation occurs at different level (presence of inducing amino acids, cytosolic interactors, etc.). In addition, only a subset of genes are affected in the *stp2* strain relative to *zcf25* strain. Unless more data is presented that dissects this relationship, it suggested that the authors tone this down.

Point-by-point response

Reviewer #1

By means of metabolic and transcriptional profiling of human pathogenic yeast *Candida albicans*, Garbe et al. uncovered and characterized a link between the metabolism of aromatic amino acids and the 3-oxoadipate pathway (3-OAP) involved in the catabolism of hydroxybenzenes. They also investigated roles of the transcription factors Stp2, Zcf25, and Zcf10 and a gene coding for a putative anthranilate 1,2-dioxygenase C4_00290C. The genes for Zcf25 and Zcf10 are present in the metabolic gene clusters (MGCs) coding for the 3-OAP enzymes and the authors demonstrated that Zcf25 controls the catechol branch of the 3-OAP as well as the shikimate pathway. The manuscript is well written, and the results appear sound and well justified.

Minor comments:

1. Lines 346-347: The authors propose a hypothesis that Zcf10 and Zcf25 are involved in the transcriptional control of the 3-OAP MGCs. However, the roles of both transcription factors in the 3-OAP regulation have been proposed about 10 years ago by Gerecova et al. 2015.

This is absolutely true and we therefore reference Gerecova et al. (2015) multiple times throughout our manuscript to highlight its central contribution to this field of research and importance for our work. In our study, we contribute, for the first time to our knowledge, experimental evidence for this hypothesis, especially for the activity of Zcf25 as the regulator of its gene cluster and the catechol-branch of the 3OAP in C. albicans. In order to clearly acknowledge the original hypothesis from the novel findings we could add with our study, the corresponding section in the results part has been changed to the following: "Together with our data-driven findings from the metabolic profiling and modeling concerning the SHKP, we inferred the following two hypotheses: (I) The SHKP and 3OAP are connected in C. albicans; (II) amino acid and hydroxybenzene metabolism are further connected on gene regulatory level, with Stp2 acting as a repressor of catechol branch of the 3OAP, encoded by the ZCF25 gene cluster. Additionally, we aimed to verify the hypothesis that Zcf10 and Zcf25 are regulators of their respective gene clusters and are required for the metabolism of hydroquinone and catechol, respectively (31)." (Lines 326-332)

2. I would suggest mentioning also the changes in expression of PHH1 and PHH2 genes encoding phenol hydroxylases that result from the knock-outs of ZCF10 and ZCF25 (e.g. in the paragraph on lines 452-469).

We thank the reviewer for this suggestion and agree with the benefit of mentioning this additional information. We included it into our discussion, which has been rephrased to the following: "Further, we also observed notable induction of the putative phenol hydroxylases PHH1 (CR_07940W) and PHH2 (CR_07820W) in response to hydroquinone. While we confirmed that Zcf10 is actually expendable for growth on hydroquinone, we observed a partially Zcf25-dependent induction of PHH2 in response to catechol, indicating Zcf25 regulatory influence on phenol catabolism." (Lines 453-457)

3. *C. albicans* assimilates phenol via the catechol branch of the 3-OAP. Although I do not insist on this, it would be great to add to Figure 5 also a panel with the strains utilizing phenol as a sole carbon source.

We acknowledge that this addition would add another facet to our manuscript. In fact, we generated additional stress tests showing that zcf25Δ is slightly more sensitive to phenol on SD/YPD base medium. We also investigated fungal growth using phenol as the sole carbon source with several C. albicans strains. Under the tested conditions, these experiments did not yield convincing results (Figure R1.3) since all tested strains showed growth and only a very slight impairment was visible for zcf25Δ. However, the current lack of resources prevented us from optimizing this experiment, as well as the phenol stress tests, regarding optimal phenol concentrations, solubility and general incubation conditions, which would be required to clarify the suspected role of zcf25Δ for carbon assimilation from phenol. Therefore, while we refrained from including these results in our manuscript, we include them in the point-by-point response, which will be available alongside the manuscript. Of note, the potential importance of Zcf25 for the catabolism of phenol is also indicated by its relevance for the expression of the putative phenol hydroxylase PHH2, which is now mentioned in our discussion in response to the reviewer's comment #2.

Figure R1.3: Growth of indicated C. albicans strains on phenol as a carbon source
C. albicans strains were tenfold serially diluted in sterile H₂O and spotted on agar plates containing only 10 mM phenol as the sole carbon source. Plate was incubated for 2 days at 37°C.

4. The authors could mention that similar to *C. albicans*, a knock-out of the OTF1 gene in *C. parapsilosis* (an ortholog of ZCF10) also exhibits only mild changes on the transcriptional activation of the corresponding MGC (Cillingova et al. 2022).

Thank you again for your comment, we agree with the benefit of highlighting this information and added the following statement to the discussion: "Notably, Zcf10 was expendable for growth on hydroquinone, a finding that aligns with a previous report showing that the deletion of the C. parapsilosis ZCF10 ortholog, OTF1, barely affects growth on hydroquinone or the expression of its respective gene cluster (59)." (Lines 445-448)

5. The expression of genes in the 3-OAP MGCs is repressed in glucose containing media (presumably mediated via Mig1/2). Could the authors clarify whether the presence of aromatic amino acids in cultivation media represses the genes in the 3-OAP MGCs?

In general, the full extent of regulatory influence on the 3OAP remains not fully understood and requires further investigation. Nevertheless, based on our results, some evidence for the hypothesis that aromatic amino acids (AAAs) might repress 3OAP activity are already present, as for example the Zcf25 MGC in response to SC (amino acid medium) is more strongly induced in stp2Δ than in the WT. Given that AAAs trigger Stp2-activity, the observed higher induction might be explained by the absence of AAA repression mediated by Stp2. However, this would also be true for other amino acids triggering Stp2. In addition, it is unclear if AAAs, especially tryptophan, can be catabolized via the catechol-branch of the 3OAP, which could result in cross-regulation of AAA and 3OAP. Thus, to test the hypothesis that AAAs might influence 3OAP gene expression, experimental verification in more defined media settings would be required, such as incubation of the fungus in catechol medium with vs. without AAAs or SC medium with vs. without AAAs. We added a remark towards this aspect in our discussion: "These potential novel regulatory activities of Stp2 require future experimental validation, exemplary in the direction if extracellular amino acids in general or specific sets, like AAAs, influence 3OAP activity." (Lines 462-464)

6. Line 443: The sentence "... C. albicans not only kept HHQ- but also gained the catechol branch of the 3OAP ..." is a bit misleading, as comparative studies (e.g. on C. subhashii by Mixao et al. 2021) indicate that a presumed ancestor of the CUG clade possessed both 3-OAP MGCs. Hence, C. albicans rather kept both ancestral clusters, while some other species (e.g. C. parapsilosis) lost this branch of 3-OAP.

We thank the reviewer for this valuable hint and changed the manuscript text to the following: "The sole fact that C. albicans kept both 3OAP MGCs present in the presumed ancestor of the serinales, while species like C. parapsilosis lost the catechol-branch, argues for an important function within its opportunistic lifestyle (31, 62)" (Lines 426-428)

Reviewer #2

This work by Garbe et al. focuses on the utilization of metabolomics and transcriptomics approach (lumped together as "multi-omics") to unravel potential mechanisms by which the human fungal pathogen *Candida albicans* is able to adapt in medium lacking extracellular amino acids (exAA), enabling the restoration of metabolic balance in the cell. The authors used established analyses pipelines to identify DEGs and altered metabolites in cells grown under specific conditions. Central to this work is the transcription factor Stp2, which has been shown by previous studies, including that of the authors, to mediate the assimilation of extracellular amino acids through the derepression of a specific subsets of amino acid transporters. The work nicely captured the outcome expected of *stp2* like its capacity to remain adaptable albeit in a very a delayed manner in AA-rich conditions. The authors found that the *stp2* KO in exAA-lacking condition, is characterized by the upregulation of shikimate pathway (aromatic AA biosynthesis) and catabolism of hydroxybenzene, the latter inferred from the upregulation of specific genes clusters linked to 3-oxoadipate (3-OAP) pathway. The 3-OAP is suggested to be controlled by another transcription factor Zcf25, which the authors suggest to be negatively regulated by Stp2.

The work provides sufficient overview and presentation of key differences among strains grown under specific conditions. However, as this work focuses on amino acid metabolic processes and all -omics interpretations were drawn from cells grown under specific conditions, it is important that the specific culture/pre-culture conditions are described and substantiated. In addition, integrating two transcriptomics and metabolomics approaches, need to meet this consistency element to make the data useful in obtaining a comprehensive understanding of biological processes.

Comments:

1. To consider the shifting of cells from complex medium like YPD to SD as an AA-rich to AA-poor adaptation (discussed in line 419-421) is rather a misnomer as ammonium, which is present in all test media, is immediately sequestered by the central nitrogen metabolism to generate key amino acids like glutamine/glutamate, which can then be converted to a plethora of other amino acids. This is nicely reflected in metabolomics data in Fig. S1F. Adding amino acids (1% SC) to the base medium would by-pass this dependency on central nitrogen metabolism, allowing other amino acids to be sequestered by other pathways. If any, the switch is a reflection of the cells need to convert AA to glucose, and not because of AA-limitation/starvation. The use of other non-preferred nitrogen source like allantoin/purines in the base medium, would be more ideal in this regard since they are not immediately converted to key amino acids.

*The reviewer addresses an important limitation in our study design. Indeed, it would be valuable to additionally examine the investigated metabolic processes also in media containing other, non-preferred nitrogen sources like allantoin. Although we do not have this data, we are confident that also in the conditions we had chosen valuable insights could be obtained. In general, it must be distinguished between the adaptation of the WT to amino acid-rich, glucose-free conditions (investigated on a global scale), which likely reflects an adaptation to conversion of specific amino acids to others and a shift to gluconeogenesis, as rightfully pointed out by the reviewer and the adaptation of *stp2Δ* to these conditions. Since it is known which amino acids will not be imported by this strain, we were able to model a specific situation for starvation of extracellular amino acids. In addition, despite the presence of ammonium in all media, we still observed a delayed adaptation of *stp2Δ*, pointing to specific metabolic costs of generally possible adaptation mechanisms. Lastly, we aimed at identifying potential, so far unknown, metabolic mechanisms beneficial for the adaptation to such conditions, like the 3OAP. In order to better reflect the limitations of our study,*

we extended the last discussion section to the following: "In conclusion, a number of questions remain unanswered, for example – Is the catechol feeding the 3OAP in C. albicans of endogenous or exogenous origin?; If exogenous, where does it come from in the host and how is it sensed and transported into the fungus?; Which enzyme(s) links the 3OAP and SHKP?; How are central regulatory circuits, like nitrogen or carbon catabolite repression, affecting these potential connections and the 3OAP in general (9)?" (Lines 494-499)

2. All throughout the text, the authors mentioned the use of "SC" as a glucose-free, amino acid-rich condition where Stp2 processing is induced (from line 124). I am unable to trace the specific description of this medium in the text other than the mention of the supplier (Biomol; line 530). Can the authors please clarify this? This is important because not all amino acids are capable of inducing Stp2 processing and understanding the proportion of available amino acids would provide an indication as to how these fluxes would operate. The authors also suggested that under the "SC" condition, the Stp2 is fully required (line 125). However, the drop plate shown in Fig. 5A, does not fully reflect this as the *stp2* had appreciable growth. Any comments on this? Have the authors checked the phenotypes in liquid cultures as well?

We apologize for not being precise with the definition of the used culture conditions, especially with media like SC, which is used in various compositions. In our study "SC" refers to a glucose-free and amino acid-rich medium (composition of all proteinogenic amino acids). To clarify this and provide the required information at a glance, we have added a sheet to Table S6 containing the used media recipes, as well as the order number for the used "SC" mix, and referenced it in our text.

*We agree with the reviewer's comment that using the term "fully" required is misleading. This was used to illustrate the difference in dependence of C. albicans on Stp2 for growth on amino acid-rich medium, where the growth impairment of *stp2Δ* is more severe on SC than CAA. Although not detailed here, we noticed substantial differences during our experiments using either of those media. Unsurprisingly, it also made a difference if liquid or solid medium was used, with typically more pronounced effects in liquid medium. Further, growth defects were also more severe in small volumes (microtiter plates vs. flasks) than in bigger volumes using similar starting ODs. Likely, at least part of these effects can probably be attributed to different oxygen availability in microtiter plates vs. flasks but were not detailed in this study. However, *stp2Δ* still showed some growth on SC when incubated in liquid SC medium in flasks as shown in Fig. S4A. In order to avoid misleading terminology, the respective manuscript section in the results section was modified as follows: "We performed global metabolomic analysis to obtain cellular metabolite profiles of C. albicans WT and *stp2Δ* cells incubated for 90 min in Stp2 non-inducing (SD, glucose-rich and amino acid-free) and inducing media (SC, glucose-free and amino acid-rich; and CAA, glucose-free, amino acid and peptide-rich) (27, 28, 30)." (Lines 117-120)*

3. It would benefit the readers to restructure the methods section particularly that concerns the transcriptional profiling/RNASeq (e.g., line 533-560) as there are several conditions/experiments described. Can the authors confirm that the preculture conditions (YPD) prior to media switch for RNASeq and metabolomics for WT/*stp2* are the same? Any specific reasons as to why in some cases the authors switched to SD (2% glucose) as pre-culture condition? The biological insights on AAA/HB metabolism were drawn from the first metabolic profiling using YPD as pre-culture.

We confirm that culture conditions of the transcriptional data set I and the metabolomics data set are the same, with the sole difference that the fungal cells have been incubated for 90 min for metabolomics and 60 min for transcriptomics, due to the assumption that transcriptional changes observed after 60 min might take a bit longer to manifest in the metabolome. For the transcriptional data set II the preculture condition were indeed changed to SD instead of YPD with the objective to constantly culture the fungus in clearly defined media conditions and vary only the carbon source. Furthermore, the incubation time for the second data set was increased to 240 min also in order to test the impact of a prolonged time frame and stability of RNA profiles. Although the transcriptional responses remained similar between both transcriptome datasets, the prolonged time frame potentially explains slight differences in the expression levels of distinct genes. For example, zcf25Δ is significantly upregulated in stp2Δ vs. WT in SC in the first data set (60 min) but misses the significance threshold in the second data set (240 min) (Fig. 3 and 5). In order to allow quick traceability of the used experimental conditions, we also added an overview to Table S6 and referenced it in our materials and methods section. Moreover, we also carefully revised our results, and clarified which dataset was used for which analysis.

4. In addition, any reasons as to why the glucose condition was changed from 2% to 10 mM in catechol RNASeq experiment (line 555-557)? This is essentially the focus of Fig. 5. Is this correct? Does the "SD" used as a comparator to assess in gene clustering have 10 mM or 2% glucose? It is well-known that glucose levels has a profound influence on metabolic activity and downstream processes, and I wonder whether the key suggestions like stp2 gene expression in catechol resembling that of stp2 in SD medium is still valid?

Indeed, the glucose concentration for SD test medium of the transcriptional data set II was set to 10 mM instead of the 2% used for the first data set and metabolomics. Therefore, "SD" in this set refers to medium with 10 mM. We also agree with the point raised by the reviewer regarding the influence of different glucose levels. In fact, we intentionally aimed to test comparable amounts of the carbon source media components in the mM range for all tested media in the second transcriptional data set (Glucose, catechol, hydroquinone and SC – the mixture of all proteinogenic amino acids) and wanted to avoid possible regulatory effects due to excess availability of glucose in comparison to the other tested carbon sources. However, compared to the initial data set the effects of STP2 deletion on the expression of the ZCF25 gene cluster were still present.

5. The idea that Stp2 represses Zcf25 is interesting. However, caution is needed in over interpreting the data in Fig. 5D as Stp2 regulation occurs at different level (presence of inducing amino acids, cytosolic interactors, etc.). In addition, only a subset of genes are affected in the stp2 strain relative to zcf25 strain. Unless more data is presented that dissect this relationship, it suggested that the authors tone this down.

We agree that the 3OAP-associated regulation patterns are complex and multi-faceted. We acknowledge it exceeded the scope of our study to dissect the influence of all possible regulators on the SHKP and 3OAP as well as on the Stp2 and Zcf25 interplay. However, we are confident to formulate this as hypothesis based on our data – exemplary upregulation of the complete Zcf25 cluster in stp2Δ in response to SC in the transcriptional data set I, partial upregulation of the cluster in transcriptional data set II, the finding that more than 400 genes downregulated in zcf25Δ in response to catechol are upregulated in stp2Δ (Fig. S4E). Nevertheless, we agree that this remains a hypothesis that necessitates experimental verification and added the following statement to the

discussion to illustrate the limits of our study: “These potential novel regulatory activities of Stp2 require future experimental validation, exemplary in the direction if extracellular amino acids in general or specific sets, like AAAs, influence 3OAP activity.” (Lines 462-464)

Re: mSystems00226-25R1 (**A multi-omics analysis unveils functional and regulatory links between hydroxybenzene and aromatic amino acid metabolism in *Candida albicans***)

Dear Dr. Enrico Garbe:

Your manuscript has been accepted, and I am forwarding it to the ASM production staff for publication. Your paper will first be checked to make sure all elements meet the technical requirements. ASM staff will contact you if anything needs to be revised before copyediting and production can begin. Otherwise, you will be notified when your proofs are ready to be viewed.

Sincerely,
Benjamin Wolfe
Editor
mSystems

Reviewer #1 (Comments for the Author):

The authors have adequately addressed all my comments.

Reviewer #2 (Comments for the Author):

I really appreciate the time and effort to address the points raised. I am satisfied with the authors' response and modifications. Thank you.